# VARIATIONAL NEURO-SYMBOLIC GENERATIVE TEMPORAL POINT PROCESS

## ABSTRACT

Temporal point processes (TPPs) are a powerful framework for modeling event sequences with irregular timestamps, such as those commonly found in electronic health records (EHR), which often involve high-dimensional and diverse event types. However, building generative models for such complex datasets comes with several challenges, including addressing sample inefficiency, accurately capturing intricate event patterns, and producing outputs that are both trustworthy and interpretable. In this paper, we present a neuro-symbolic generative model for TPPs based on the Variational Autoencoder (VAE) framework. Our model incorporates a neural-symbolic reasoning layer into the latent space, allowing it to integrate interpretable, logic-based constraints and perform logical reasoning over learned representations. This integration enhances the interpretability of the latent space by embedding logic rules directly into the generative process, enabling structured reasoning and improved decision-making based on underlying data patterns. We validate our model on an ICU EHR dataset, as well as other real-world datasets, demonstrating its effectiveness in capturing complex event dynamics with irregular timestamps. In addition to improving sample efficiency and accuracy, our model supports the secure and interpretable generation of synthetic event data, making it a valuable tool for healthcare applications where reliability and trustworthiness are critical.

## 1 INTRODUCTION

Temporal point processes (Daley & Vere-Jones, 2007) are a powerful model for analyzing and predicting event sequences occurring at irregular time intervals. These models are particularly well-suited for capturing the dynamics of events in continuous time, allowing for a flexible representation of temporal dependencies and event relationships. TPPs are widely applied in various domains, from finance and social media to healthcare (Reynaud-Bouret & Schbath, 2010; Bacry et al., 2015; Zhao et al., 2015; Farajtabar et al., 2017), where understanding the generative process of the event time and types plays a crucial role in understanding underlying patterns and making predictions.

For example, in healthcare, TPPs offer a valuable tool for modeling EHRs (Enguehard et al., 2020), which contain detailed sequences of medical events such as drug prescriptions, diagnoses, and monitoring of vital signs. These sequences are often characterized by irregular time intervals and complex dependencies, making them challenging to model effectively. In this context, generative models hold significant value for several reasons. First, they provide a powerful tool for augmenting limited real datasets, particularly in cases of rare diseases where patient data is scarce (Lee et al., 2022). By generating synthetic data that adheres to clinical logic and medical guidelines, these generative models can facilitate more robust analysis and support the development of treatments or interventions. Second, generative models can help create secure, de-identified patient data (Libbi et al., 2021; Biswas & Talukdar, 2024; Biswal et al., 2021), ensuring privacy while enabling researchers and clinicians to explore new insights without compromising sensitive information. Additionally, synthetic data can be used to simulate various medical scenarios, aiding in the testing of predictive models and decision-support systems.

Traditional generative models, such as VAEs (Kingma, 2013), are highly flexible and effective at learning complex distributions. However, they often fall short in providing interpretable latent representations that align with clinical logic. This lack of interpretability can limit their ability to gener-

ate realistic and clinically relevant data, which is essential for high-stakes applications in healthcare where trustworthiness and alignment with medical knowledge are critical.

To address these challenges, we propose a neuro-symbolic generative model for TPPs, leveraging the VAE framework to integrate interpretable logic-based constraints into the latent space. Our model incorporates a neuro-symbolic reasoning layer that enhances the VAE's capabilities by embedding domain knowledge directly into the generative process. This layer utilizes predicate embeddings from the encoder, which represent abstracted event information. Through forward chaining (Campero et al., 2018; Glanois et al., 2022)—a logical inference process—the model refines these embeddings to infer consistent states, which are then used by the decoder to generate event sequences.

This approach not only improves the interpretability of the latent representations but also ensures that the generated synthetic data adheres to clinical standards and reflects real-world medical logic. By applying our model to EHR datasets, particularly for rare diseases, we can generate semi-synthetic data that is both realistic and useful for research purposes. This capability is crucial for enhancing data privacy, testing clinical hypotheses, and developing treatment strategies, especially in scenarios where real data is limited.

**Contributions**   Our specific contributions include: *i)* Incorporate neuro-symbolic reasoning layers into the VAE framework to significantly enhance the interpretability of the latent representation. *ii)* Accomplishing the challenging temporal point process generation in practice, which is important for generating de-identified data and handling missing data. Well-trained models can leverage domain knowledge to generate semi-synthetic datasets from actual data, aiding in transfer learning and secure data generation for privacy protection. *iii)* Compressive experiments with real-world datasets demonstrate that mined rules not only align with real-world scenarios, but also prove advantageous for both prediction and generation tasks.

## 2 RELATED WORK

**Temporal Point Processes (TPP)**   TPP models have emerged as an elegant framework for modeling event times and types in continuous time, directly treating the inter-event times as random variables. With the advance of neural network, various neural TPP models have been proposed. Some of them are built on recurrent neural networks (Du et al., 2016; Mei & Eisner, 2017; Xiao et al., 2017b; Omi et al., 2019; Shchur et al., 2019; Mei et al., 2020; Boyd et al., 2020). Some others utilize transformer architecture (Zuo et al., 2020; Zhang et al., 2020; Enguehard et al., 2020; Sharma et al., 2021; Zhu et al., 2021; Yang et al., 2021). Recently, Li et al. (2020; 2021) proposed integrating logic rules within the intensity function of TPP model to foster interpretability. *We aim to utilize TPP to model the real-world event sequences.*

**Variational Auto-Encoder (VAE)**   VAE models, proposed by Kingma (2013), encodes data to latent (random) variables, and then decodes the latent variables to reconstruct the input data. Recent works resort to VAE to learn a disentangled representation for sequential data. Bowman et al. (2015) succeed in training a sequence-to-sequence VAE and generating sentences from a continuous latent space. Desai et al. (2021) proposed a VAE framework with a decoder design that enables user-defined distributions for generating time-series data. To enhance the disentanglement capability, some works aim to introduce structural patterns in latent representation of VAE to improve efficiency and the quality of generated data. Hu & Rostami (2023) proposed a binarized regularization for VAE to encourage symmetric disentanglement, improve reconstruction quality. Van Den Oord et al. (2017) incorporated ideas from vector quantisation to learn a discrete latent representation which model important features that usually span many dimensions in data space. Yang et al. (2020) proposed a new VAE framework which includes a causal layer to transform independent exogenous factors into causal endogenous ones that correspond to causally related concepts in data. *However, existing VAE frameworks often lack interpretability in the latent representation, overlooking fine-grained guiding logic rules.*

**Neuro-Symbolic Integration**   Neuro-Symbolic systems aims to transfer principles and mechanisms between logic-based computation and neural computation Besold et al. (2021). Serafini & Garcez (2016) demonstrate that the logic can be implemented using neural networks for the groundings of the symbols. Manhaeve et al. (2018) started from a probabilistic logic programming language

and extended it to handle neural predicates. Kusters et al. (2022) proposed a neural architecture that learns literals that represent a linear relationship among numerical input features along with the rules that use them. Campero et al. (2018) proposed a neuro-symbolic approach, in the sense that the rule predicates and core facts are given dense vector representation, for logical theory acquisition. Glanois et al. (2022) proposed a neuro-symbolic model to solve inductive logic programming problems. Recently, Yang et al. (2024) introduced a neuro-symbolic rule induction framework within the temporal point process model. *We aim to integrate neuro-symbolic module in VAE framework for pattern mining in the latent representation, thereby improving interpretability and enabling the generation of sequences in an explainable manner.*

## 3 BACKGROUND

### 3.1 MULTIVARIATE TEMPORAL POINT PROCESSES (MTPPS)

MTPPs provide a mathematical framework for modeling sequences of events over time, where each event is characterized by a timestamp $t_i \in \mathbb{R}^+$ and an event type (or marker) $m_i \in \mathcal{M}$. The event timings $\{t_i\}$ are irregular, with intervals $\Delta t_i = t_{i+1} - t_i$ governed by an underlying stochastic process, typically modeled by a conditional intensity function.

The *conditional intensity function* $\lambda_k(t \mid \mathcal{H}(t))$ represents the instantaneous rate at which events of type $k \in \mathcal{M}$ occur at time $t$, given the history $\mathcal{H}(t)$ of all past events:

$$\lambda_k(t \mid \mathcal{H}(t)) = \lim_{\Delta t \to 0} \frac{\mathbb{P}(\text{ event of type } k \text{ occurs in } [t, t + \Delta t) \mid \mathcal{H}(t))}{\Delta t}$$

where $\mathcal{H}(t)$ denotes the set of all events $(t_i, m_i)$ that occurred prior to time $t$.

Given an observed sequence of events $\{(t_i, m_i)\}_{i=1}^N$, the likelihood of this sequence under an MTPP model is:

$$L\left(\{(t_i, m_i)\}_{i=1}^N\right) = \left(\prod_{i=1}^N \lambda_{m_i}(t_i \mid \mathcal{H}(t_i))\right) \exp\left(-\int_0^T \sum_{k=1}^K \lambda_k(t \mid \mathcal{H}(t))dt\right)$$

Maximizing this likelihood (or its log-likelihood) is the standard approach for estimating the parameters of an MTPP model, providing a principled way to model event data with irregular intervals and multiple event types.

### 3.2 RULE LEARNING AND REASONING

We consider learning logic rules in the form of Horn clauses:

$$f: \quad Q \leftarrow P_1 \wedge P_2 \wedge \cdots \wedge P_h \tag{1}$$

Here, $P_1, P_2, \ldots, P_h$ are predicates that form the *body* of the rule, representing conditions that must hold true, and $Q$ is the *head*, representing the conclusion that can be inferred when all the predicates in the body are satisfied.

Predicates are Boolean variables that take values of either True or False, based on the data. They express properties or relationships between entities. For example, a predicate like *HasFever(Patient)* indicates whether a patient has a fever, while *UseDrug(Patient)* specifies whether a particular drug is being administered. These predicates help capture key characteristics and relationships within the system's state.

With learned rules, we can reason to infer new information. Reasoning includes two main methods: *forward chaining*, which starts from known facts (body predicates) and infers new knowledge step by step, and *backward chaining*, which starts from a goal (the head) and works backward to find supporting conditions. In this work, we focus on forward chaining, which is especially useful for predicting future events in temporal point processes (TPP).

For example, given the rule *UseDrug ← HasFever ∧ HighTemperature* and knowing that *HasFever(Patient)* and *HighTemperature(Patient)* are true, we can infer via forward chaining that *UseDrug(Patient)* is true.

### 3.3 Variational Autoencoder (VAE)

In our setting, we will extend the VAE framework for sequential event generation. Given a dataset $X$ consisting of event sequences $\boldsymbol{x} = \{(t_i, m_i)\}_{i=1}^N$, where $t_i$ represents the timestamp and $m_i$ the event type (marker), our goal is to generate new sequences of events by modeling the unknown joint probability distribution $p(\boldsymbol{x})$. Specifically, at each step, we model the conditional distribution of the next event $(t_{i+1}, m_{i+1})$ given the latent variable $z$ and the history of prior events $\mathcal{H}(t_i) = \{(t_j, m_j)\}_{j=1}^i$.

In our VAE framework, the generative process works as follows: First, the *encoder* maps an observed event sequence $\boldsymbol{x}$ to a approximate posterior distribution $q_\phi(\boldsymbol{z} \mid \boldsymbol{x})$, where a latent variable $\boldsymbol{z}$ is sampled, representing initial understanding of the global structure of the event sequence. Second, the initial latent variable $\boldsymbol{z}$ is then passed through a learnable *Neuro-Symbolic forward reasoning* module, where forward reasoning updates the state of certain components of $\boldsymbol{z}$. This module will be explained in detail later. Last, the *decoder* auto-regressively models the conditional distribution of the next event given the latent variable $\boldsymbol{z}$ and the current history $\mathcal{H}(t_i)$, i.e., $p_\psi(t_{i+1}, m_{i+1} \mid \boldsymbol{z}, \mathcal{H}(t_i))$. This autoregressive process repeats until a complete sequence is formed.

The loss function for this VAE framework is again based on the Evidence Lower Bound (ELBO) loss function, but it now accounts for the auto-regressive nature of the process:

$$L_{\psi,\phi} = -\mathbb{E}_{q_\phi(\boldsymbol{z}|\boldsymbol{x})}\left[\sum_{i=1}^N \log p_\psi(t_i, m_i \mid \boldsymbol{z}, \mathcal{H}(t_{i-1}))\right] + D_{KL}\left[q_\phi(\boldsymbol{z} \mid \boldsymbol{x})\|p_\psi(\boldsymbol{z})\right] \quad (2)$$

Here, the first term ensures that the decoder learns to generate realistic event sequences by sequential conditioning on both the latent variable $\boldsymbol{z}$ and the event history; the second term regularizes the approximate posterior distribution $q_\phi(\boldsymbol{z} \mid \boldsymbol{x})$ to match the prior $p_\psi(\boldsymbol{z})$, preventing over-fitting.

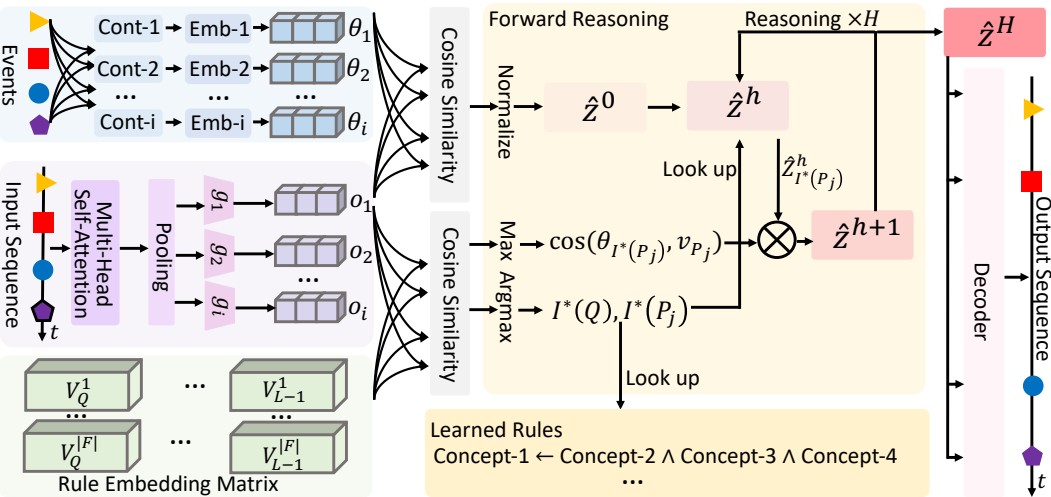

Figure 1: Model Framework.

## 4 Model: Variational Neuro-Symbolic Generative TPP

Our model framework, depicted in Fig.1, begins with the introduction of latent states, embedding representations of predicates, and logic rules, followed by encoder-decoder modules and the learning process, which will be illustrated in detail in following subsections.

### 4.1 Latent State, Embedding Representation of Predicates, and Logic Rules

We introduce a latent state variable $\boldsymbol{z} \in \{0, 1\}^d$, where each of the $d$-dimensional components represents a binary Boolean variable. Each element of $\boldsymbol{z}$ represents the satisfaction of a specific concept

or grounded predicate, indicating whether a condition is true or false based on the data. The concepts or predicates are extracted by the original event, which are high-level and concise. We will, therefore, have $d$ concepts or predicates. In the context of healthcare, these concepts or predicates could represent critical medical concepts such as *HighBloodGlucose*, *IrregularHeartbeat*, or higher-level concepts such as *MedicationAdherence*, *HospitalizationRisk*, and so on. Each element $z_i$ represents the inferred predicate states.

Note that $\boldsymbol{z}$ can guide the generative process of future events. For example, if $z_1$, such as *HasFever* has an inferred value close to 1 would influence the model to predict follow-up medical interventions, such as prescribing antipyretic medication, while the inferred $z_1$ near 0 would suggest that no fever is present, thus shifting the model's predictions accordingly.

We assume that each element $z_i \in \boldsymbol{z}$ (where $\boldsymbol{z} \in [0,1]^d$ ) follows a Bernoulli prior with a probability $p_i$, which means each $z_i$ is sampled from a Bernoulli distribution:

$$z_i \sim \text{Bernoulli}\,(p_i) \tag{3}$$

The prior over the latent variable $\boldsymbol{z}$ is:

$$p(\boldsymbol{z}) = \prod_{i=1}^{d} \text{Bernoulli}\,(z_i \mid p_i) \tag{4}$$

where each $z_i$ represents a binary concept that could be present or absent.

For all the pre-specified $d$ concepts or predicates, we embed each predicate as a vector $\theta_i \in \mathbb{R}^k$ with dimension $k$, for $i \in \{1, 2, \ldots, d\}$. Each corresponds to a distinct medical concept or predicate. These predicates are embedded as continuous vectors. In this embedding space, each predicate $\theta_i$ is mapped to a vector that captures not only its individual meaning but also its relationships to other predicates. The predicate embedding set is denoted as $\Theta = \{\theta_i\}_{i=1,\ldots,d}$, which can be get from pre-training like (Mikolov et al., 2013).

These predicates are used to construct logical rules in the form of Horn clauses that describe medical diagnoses and treatment protocols. To integrate these logic rules into our generative model to aid reasoning in the latent space, we utilize a neuro-symbolic reasoning approach Campero et al. (2018); Glanois et al. (2022). Specifically, for each rule, we construct a *rule embedding matrix* $V_f \in \mathbb{R}^{k \times L}$ where $f \in \mathcal{F}$, each $f$ has a general form as shown in Eq.(1) and $\mathcal{F}$ is the set of all rules. Each matrix $V_f$ has dimensions $k \times L$, where $k$ is the dimension of the predicate embedding and $L$ is the maximal length of the rule (i.e., the number of predicates in the rule, including the head). The rule embedding matrix $V_f$ is defined as:

$$V_f = [\ v_Q \quad v_{P_1} \quad v_{P_2} \quad \cdots \quad v_{P_{L-1}} \ ] \in \mathbb{R}^{k \times L} \tag{5}$$

where $v_Q \in \mathbb{R}^k$ represents the head predicate $Q$ embedding, and $v_{P_1}, v_{P_2}, \ldots, v_{P_{L-1}} \in \mathbb{R}^k$ indicate the body $P_1, P_2, \ldots, P_{L-1}$ embeddings.

When expert knowledge is available in the form of logic, we just need to initialize and freeze the rule embedding by concatenating the corresponding predicate embeddings, such as

$$V_f = [\ \theta_Q \quad \theta_{P_1} \quad \theta_{P_2} \quad \cdots \quad \theta_{P_{L-1}} \ ] \in \mathbb{R}^{k \times L} \tag{6}$$

When (some of) the rules are unknown or incomplete, which is common, we can still learn new rules directly from data. Specifically, we will treat $V_f$ as model parameters. These unknown rule embeddings will be optimized during training to automatically discover meaningful rules by learning to align with predicate embeddings.

## 4.2 ENCODER

The encoder transforms an input sequence of events $\boldsymbol{x}$ into a latent representation by combining neural sequence modeling with symbolic reasoning. The process involves embedding the input sequence using a Transformer-based architecture and refining the latent variables using a *neural symbolic reasoning layer*.

Given an input sequence of events $\boldsymbol{x} = \{(t_i, m_i)\}_{i=1}^{N}$, where $t_i$ denotes the time and $m_i$ the marker of each event, we first encode this sequence using a Transformer (Zuo et al., 2020). The Transformer

outputs a sequence embedding matrix $\boldsymbol{E} \in \mathbb{R}^{N \times d_e}$, where $d_e$ is the dimensionality of the event embeddings. To obtain a single global sequence embedding, we pool the event embeddings into $\boldsymbol{e}_{\text{seq}} \in \mathbb{R}^{d_e}$ using a pooling technique such as mean pooling or attention pooling:

$$\boldsymbol{e}_{\text{seq}} = \text{Pool}(\boldsymbol{E}) \tag{7}$$

This global sequence embedding captures temporal and event-type dependencies in the input.

Next, the sequence embedding $\boldsymbol{e}_{\text{seq}}$ is fed into multiple Multi-Layer Perceptrons (MLPs) denoted as $g_1(\cdot), g_2(\cdot), \ldots, g_d(\cdot)$. Each MLP $g_i$ takes the global sequence embedding $\boldsymbol{e}_{\text{seq}}$ as input and outputs a vector of the same dimensionality as the predicate embedding $\theta_i$, i.e., $\in \mathbb{R}^k$, which represented as:

$$\boldsymbol{o}_i = g_i\left(\boldsymbol{e}_{\text{seq}}\right) \in \mathbb{R}^k \tag{8}$$

Then we compute the similarity between the MLP output $\boldsymbol{o}_i \in \mathbb{R}^k$ and the corresponding predicate embedding $\theta_i \in \mathbb{R}^k$ using cosine similarity, which yields initial inference. To ensure the output is in the range $[0, 1]$, we normalize the cosine similarity:

$$\hat{z}_i^{(0)} = \frac{1 + \text{CosineSimilarity}\left(\boldsymbol{o}_i, \theta_i\right)}{2}, \quad \hat{z}_i^{(0)} \in [0, 1], \quad i = 1, \ldots, d \tag{9}$$

This formulation is conceptually similar to a "concept bottleneck" in VAE models, where high-level concepts serve as intermediaries for prediction. However, unlike traditional bottleneck methods that rely on a pre-trained supervised model to map inputs to predefined concepts using explicit labels (Oikarinen et al., 2023), we directly learn the encoder to align with the latent predicates. The initial guess of each concept's satisfaction is determined by the cosine similarity between the MLP output and the predicate embedding vector, which will be further refined using the *neural-symbolic reasoning layer*. This design removes the reliance on pre-training and allows the model to learn the latent concepts (predicates) dynamically based on data alignment during training.

## 4.3 NEURAL-SYMBOLIC REASONING LAYER

Given the current rule embedding $V_{\mathcal{F}} = \{V_f\}_{f \in \mathcal{F}}$, which are the model parameters, let's first assume that all rules need to be learned. The reasoning process iteratively updates the latent variables $\boldsymbol{z} \in [0, 1]^d$ by applying the learned logic rules recursively over $H$ iterations. This mimics how humans perform forward reasoning, progressively applying rules to infer new knowledge.

$$\hat{\boldsymbol{z}}^{(h+1)} = \text{Forward-Reasoning}\left(\hat{\boldsymbol{z}}^{(h)}, \Theta_{\mathcal{F}}\right) \in [0, 1]^d, \quad h = 0, 1, \ldots, H - 1 \tag{10}$$

where $h$ is the index of the forward chaining iteration. In each iteration, symbolic reasoning propagates values from the body predicates to the head predicates of each rule, mimicking human reasoning by repeatedly applying known rules to update the inferred latent variables. This process ensures alignment with the underlying logic. After $H$ iterations, the neural symbolic layer produces the updated posterior probabilities of the latent predicate variables, denoted as $\hat{\boldsymbol{z}}^{(H)} = [\hat{z}_i^{(H)}] \in [0, 1]^d$.

We now proceed to detail the architecture of the **Forward-Reasoning**$(\cdot)$ operator. Given the rule embedding matrix $V_{\mathcal{F}} = \{V_f\}_{f \in \mathcal{F}}$, where each rule $V_f$ is represented as $V_f = \left[v_Q, v_{P_1}, \ldots, v_{P_{L-1}}\right]$, we aim to iteratively update the latent variable vector $\hat{\boldsymbol{z}} \in [0, 1]^d$ by performing forward reasoning in $H$ iterations.

For each iteration, perform following steps:

**Step 1 – Determine Head and Body Predicate Indices** For each rule $f \in \mathcal{F}$, compute the indices of the head predicate $Q$ and the body predicates $P_1, \ldots, P_{L-1}$ by maximizing the cosine similarity between the predicate embedding vectors $\theta_i \in \mathbb{R}^k$ and the corresponding rule embedding vectors $v_Q, v_{P_1}, \ldots, v_{P_{L-1}}$:

$$I^*(Q) := \underset{i \in \{1, \ldots, d\}}{\arg\max} \cos\left(\theta_i, v_Q\right) \tag{11}$$

$$I^*\left(P_j\right) := \underset{i \in \{1, \ldots, d\}}{\arg\max} \cos\left(\theta_i, v_{P_j}\right), \quad \forall j = 1, \ldots, L - 1 \tag{12}$$

Here, $I^*(Q)$ is the index of the head predicate, and $I^*\left(P_j\right)$ are the indices of the body predicates that maximize the cosine similarity with the corresponding rule embeddings.

**Step 2 – Update Latent Variables** First, we consider the *intermediate variable*. For each rule $f$, compute an intermediate variable $\hat{z}^f \in \mathbb{R}^d$, which contains only one nonzero element at the index corresponding to the head predicate $I^*(Q)$:

$$\hat{z}^f_{I^*(Q)} := \prod_{j=1}^{L-1} \left( \cos \left( \theta_{I^*(P_j)}, v_{P_j} \right) \cdot \hat{z}^{(h)}_{I^*(P_j)} \right) \tag{13}$$

All other elements of $\hat{z}^f$ are set to zero.

Second, we consider *matrix formation*. We concatenate the intermediate vectors $\hat{z}^f$ for all $f \in \mathcal{F}$ into a matrix $\hat{Z}^{(h+1)} \in \mathbb{R}^{d \times |\mathcal{F}|}$, where each column corresponds to the intermediate update from a specific rule. Last, we consider the row-wise maximum. To obtain the final update $\hat{z}^{(h+1)} \in \mathbb{R}^d$, apply the maximum operation row-wise across the matrix $\hat{Z}^{(h+1)}$ :

$$\hat{z}^{(h+1)}_i = \max_{f \in \mathcal{F}} \hat{Z}^{(h+1)}_{i,f}, \quad i = 1, \ldots, d \tag{14}$$

This ensures that for each predicate $i$, the most confident rule application is selected for updating the latent variable.

$$\hat{z}^f_{I^*(Q)} := \prod_{j=1}^{L-1} \left( \cos \left( \theta_{I^*(P_j)}, v_{P_j} \right) \cdot \hat{z}^{(h)}(I^*(P_j)) \right) \tag{15}$$

All other elements of $\hat{z}^{(h+1)}_f(\cdot)$ are set to zero, meaning only the entry corresponding to the head predicate is updated.

**Step 3 – Repeat the above iteration $H$ steps** We finally get $\hat{z}^{(H)}$, which is the posterior probability of the binary latent predicates $z$.

## 4.4 DECODER

During the reconstruction phase, we derive the inferred $\hat{z}^{(H)}$ from the neural-symbolic layer and sample the latent variable $z$ from Bernoulli distribution. The decoder then auto-regressively models the conditional distribution of the next event based on $z$ and the current event history $\mathcal{H}(t_i) = p_\psi(t_{i+1}, m_{i+1} \mid z, \mathcal{H}(t_i))$ until a complete sequence is generated.

In the generation phase, after the model being well-trained, we start from an initial state and sample $z$ from inferred $\hat{z}^{(H)}$. These are input into a feed-forward neural network to construct intensity, from which inter-event times are sampled. This process is iterated, with each generated event becoming part of the historical events along with the sampled $z$, until a predefined time horizon is reached.

## 4.5 LEAERNING

The objective function for our proposed framework is based on the Evidence Lower Bound (ELBO), which now accounts for the auto-regressive nature of the process:

$$L_{\psi,\phi} = -\mathbb{E}_{q_\phi(z|x)} \left[ \sum_{i=1}^{N} \log p_\psi(t_i, m_i \mid z, \mathcal{H}(t_{i-1})) \right] + D_{KL} \left[ q_\phi(z \mid x) \| p_\psi(z) \right] \tag{16}$$

The reconstruction term is given by the summation of intermediate likelihood of multivariate point processes. To compute the KL divergence between two $d$-dimensional Bernoulli distributions where $p_\psi(z) = (p_1, p_2, \ldots, p_d)$ is the Bernoulli distribution with parameters $p_i$. And $q_\phi(z \mid x) = \left( \hat{z}^{(H)}_1, \hat{z}^{(H)}_2, \ldots, \hat{z}^{(H)}_d \right)$ is the Bernoulli distribution with parameters $\hat{z}^{(H)}_i$.

Therefore, the KL divergence $D_{KL}\left[ q_\phi(z \mid x) \| p_\psi(z) \right]$ is given by:

$$D_{KL}\left[ q_\phi(z \mid x) \| p_\psi(z) \right] = \sum_{i=1}^{d} \left[ \hat{z}^{(H)}_i \log \left( \frac{\hat{z}^{(H)}_i}{p_i} \right) + \left( 1 - \hat{z}^{(H)}_i \right) \log \left( \frac{1 - \hat{z}^{(H)}_i}{1 - p_i} \right) \right] \tag{17}$$

## 5 EXPERIMENTS

### 5.1 EXPERIMENTAL SETUP

To evaluate the effectiveness of our proposed framework, we primarily compare the model performance on prediction and generation tasks. The results indicate that our model outperforms other existing methods in prediction accuracy and demonstrates higher data generation quality. Furthermore, we visualize the process of neuro-symbolic forward chaining, which highly enhances the interpretability.

**Datasets** We utilized four real-world datasets: *i) MIMIC-IV*: An electronic health record dataset of ICU patients, focusing on those diagnosed with sepsis (Saria, 2018). We extracted 2000 samples, with an average sequence length of 22.93 events, including lab measurements, drug intake, and other health-related features. *ii) Covid-19 UK*: Collected data from the Oxford Covid-19 Government Response Tracker (Hale et al., 2021), focusing on the UK during 2021. This dataset includes 27 samples with an average of 59.22 events per sequence, tracking government policies and their impact on confirmed case reduction. *iii) Car-Follow*: Derived from the Lyft Level-5 dataset (Li et al., 2023), containing 5000 samples with an average of 4.6 events per sequence, focusing on vehicle driving modes. *iv) Epic-Kitchen*: A dataset of first-person recordings from kitchen activities, where we extracted 400 samples with an average sequence length of 36.76 events, focusing on cooking-related action verbs. For detailed descriptions and processing information of the datasets, please refer to Appendix.A.

We abstract the features in each dataset into high-level concepts and use these concepts to construct ground truth governing logic rules either by experts or large language models (Zhao et al., 2023), with details illustrated in Appendix.B.

**Baselines** We choose several state-of-the-art baselines considering three different fields: *i) Neural Temporal Point Process Model (Neural TPP)*: RMTPP (Du et al., 2016), THP (Zuo et al., 2020), PromptTPP (Xue et al., 2023), and HYPRO (Xue et al., 2022) *ii) Logic-Based Model*: TELLER (Li et al., 2021), and CLNN (Yan et al., 2023) *iii) Generative Model*: We follow the work of (Lin et al., 2022) and consider history encoder and probabilistic decoder framework for temporal point process generative model. For the history encoder, we use attention mechanism Vaswani (2017); Zuo et al. (2020). For the generative probabilistic decoder, we consider TCDDM (Sohl-Dickstein et al., 2015), TCVAE (Pan et al., 2020), TCGAN (Xiao et al., 2017a), and TCCNF (Mehrasa et al., 2019). These generative models can also be utilized for prediction tasks (Lin et al., 2022). Detailed introduction for the baselines can be found in Appendix.C

**Comparison Metric** The evaluation metrics we utilized primarily encompass the following two aspects: *i) Prediction tasks*: Following common next-event prediction task in TPPs (Du et al., 2016; Zuo et al., 2020), our model as well as other baselines (including all Neural TPP, Logic-based, and generative baselines) attempt to predict next event from history. We evaluate the event type prediction with the Error Rate (ER%) and evaluate the event time prediction with the Root Mean Square Error (RMSE). *ii) Generation tasks*: To assess the quality of the generated data, we train a classification model to distinguish between the original and synthetic data as a supervised task. Therefore, we can use the discriminative score, which is given by *(accuracy - 0.5)* on the held-out set, to evaluate the generative performance. A score close to 0 is better, indicating the generated data is hard to distinguish from original data. We also analyse 2-dimensional t-SNE plots of the original and generated data. Initially, we employ the same embedding to project the sequences into a high-dimensional space, considering that the sequences encompass both complex event time and event type information. Subsequently, t-SNE is applied to reduce the dimensionality to two components. These comparison metrics were all considered in (Yoon et al., 2019; Desai et al., 2021).

### 5.2 EXPERIMENTS FOR PREDICTION TASKS

For each dataset we mention in the experimental setup, we conduct the experiments to predict the next event. The experimental results are shown in Tab.1. Our model outperforms all generative model baselines and consistently matches or surpasses state-of-the-art neural TPP and logic-based models. Across the MIMIC-IV, Car-Following, and Epic-Kitchen datasets, our model achieves the

highest performance. Although our model ranks second on the Covid-19 dataset, it closely rivals HYPRO, the top performer, with notably stable results indicated by low standard deviation, detailed in Appendix D.

| Category | Model | MIMIC-IV | | Covid-19 | | Car-Follow | | EPIC-Kitchen | |
|---|---|---|---|---|---|---|---|---|---|
| | | ER%↓ | MAE↓ | ER%↓ | MAE↓ | ER%↓ | MAE↓ | ER%↓ | MAE↓ |
| Neural TPP | RMTPP | 92.12% | 3.75 | 62.57% | 3.52 | 36.27% | 2.64 | 42.84% | 9.21 |
| | THP | 90.38% | 3.52 | 60.74% | 3.20 | 34.70% | 2.30 | 40.25% | 9.05 |
| | PromptTPP | 86.23% | 3.27 | 54.80% | 2.95 | 34.56% | 2.10 | 37.50% | 7.80 |
| | HYPRO | 86.87% | 3.20 | **49.10**% | **2.58** | 34.35% | 2.23 | 38.25% | 8.12 |
| Logic Model | TELLER | 88.85% | 3.54 | 58.90% | 3.02 | 40.25% | 3.41 | 41.23% | 8.83 |
| | CLNN | 87.43% | 3.48 | 57.86% | 2.87 | 39.75% | 3.35 | 40.85% | 8.30 |
| Gen. Model | TCDDM | 87.58% | 3.36 | 58.23% | 3.31 | 35.38% | 2.32 | 45.34% | 8.34 |
| | TCVAE | 86.67% | 3.40 | 59.34% | 3.02 | 37.76% | 2.48 | 37.10% | 7.87 |
| | TCGAN | 85.97% | 3.29 | 58.02% | 3.12 | 34.20% | 2.58 | 39.83% | 8.20 |
| | TCCNF | 91.20% | 3.76 | 60.10% | 3.25 | 40.29% | 2.80 | 46.83% | 9.28 |
| | **Ours*** | **85.59**% | **3.13** | 53.68% | 2.74 | **33.26**% | **1.92** | **36.16**% | **7.20** |

Table 1: Comparison between our model and baselines for prediction tasks. Bold text represents the best result. The performance is averaged over three different seeds. The standard deviation can be found in Appendix.D

## 5.3 EXPERIMENTS FOR GENERATION TASKS

| Model | % Train | MIMIC-IV | Covid-19 | Car-Follow | EPIC-Kitchen |
|---|---|---|---|---|---|
| TCDDM | 100 | 0.430 +/- 0.015 | 0.489 +/- 0.003 | 0.125 +/- 0.080 | 0.368 +/- 0.028 |
| TCVAE | | 0.397 +/- 0.008 | 0.450 +/- 0.005 | 0.092 +/- 0.031 | 0.420 +/- 0.020 |
| TCGAN | | 0.388 +/- 0.007 | 0.466 +/- 0.010 | 0.100 +/- 0.023 | 0.376 +/- 0.033 |
| TCCNF | | 0.453 +/- 0.010 | 0.490 +/- 0.005 | 0.108 +/- 0.042 | 0.405 +/- 0.083 |
| **Ours*** | | **0.302 +/- 0.012** | **0.420 +/- 0.006** | **0.083 +/- 0.021** | **0.312 +/- 0.015** |
| TCDDM | 80 | 0.463 +/- 0.006 | 0.490 +/- 0.002 | 0.176 +/- 0.008 | 0.380 +/- 0.012 |
| TCVAE | | 0.433 +/- 0.010 | **0.438 +/- 0.006** | **0.130 +/- 0.008** | 0.358 +/- 0.008 |
| TCGAN | | 0.420 +/- 0.012 | 0.443 +/- 0.005 | 0.142 +/- 0.010 | 0.366 +/- 0.068 |
| TCCNF | | 0.475 +/- 0.008 | 0.492 +/- 0.002 | 0.210 +/- 0.015 | 0.402 +/- 0.099 |
| **Ours*** | | **0.382 +/- 0.006** | 0.452 +/- 0.008 | 0.132 +/- 0.016 | **0.320 +/- 0.013** |
| TCDDM | 60 | 0.465 +/- 0.008 | 0.493 +/- 0.005 | 0.232 +/- 0.012 | 0.437 +/- 0.023 |
| TCVAE | | 0.430 +/- 0.010 | 0.472 +/- 0.008 | 0.160 +/- 0.004 | 0.343 +/- 0.009 |
| TCGAN | | 0.445 +/- 0.018 | 0.475 +/- 0.005 | **0.153 +/- 0.009** | 0.352 +/- 0.016 |
| TCCNF | | 0.473 +/- 0.016 | 0.480 +/- 0.010 | 0.249 +/- 0.027 | 0.452 +/- 0.032 |
| **Ours*** | | **0.395 +/- 0.083** | **0.458 +/- 0.008** | 0.158 +/- 0.020 | **0.335 +/- 0.009** |
| TCDDM | 40 | 0.482 +/- 0.010 | 0.492 +/- 0.008 | 0.276 +/- 0.013 | 0.450 +/- 0.083 |
| TCVAE | | 0.487 +/- 0.020 | 0.490 +/- 0.008 | 0.222 +/- 0.045 | 0.431 +/- 0.037 |
| TCGAN | | 0.475 +/- 0.012 | 0.494 +/- 0.003 | 0.238 +/- 0.033 | 0.374 +/- 0.115 |
| TCCNF | | 0.492 +/- 0.006 | 0.495 +/- 0.001 | 0.430 +/- 0.020 | 0.463 +/- 0.089 |
| **Ours*** | | **0.430 +/- 0.031** | **0.479 +/- 0.013** | **0.204 +/- 0.010** | **0.362 +/- 0.012** |

Table 2: Discriminator scores for all data set, models, and training percentages. N/A's exist when not enough data was available for the model to generate synthetic data. Bold text represents the best result. The performance is averaged over three different seeds and the standard deviation is stored after "$+/-$".

**Generative Performance Evaluated by Discriminator Scores** For each dataset mentioned in the experimental setup, we use $100\%$, $80\%$, $60\%$, and $40\%$ as training data for each method. The synthetic data generated by these trained models is then used to train the post-hoc sequence classification models (by optimizing a 2-layer LSTM) to distinguish between sequences from the original and generated datasets. First, each original sequence is labeled "real", and each generated sequence is labeled "not real". Then, an off-the-shelf classifier is trained to distinguish between the two classes as a standard supervised task. Therefore, we obtain the discrimination scores, with the results shown in Tab.2.

When utilizing the complete $100\%$ training data, our model produces the best results on all datasets. Notably, all generators exhibit poor performance on the MIMIC-IV dataset due to its extensive 27 features but our model generate satisfactory sequences. At lower training proportions of $80\%$ and $60\%$, our model continues to generate superior sequences for the MIMIC-IV and Epic-Kitchen datasets. For the Covid-19 and Car-Following datasets, our model yields the second-best performance, but outperforming TCDDM and TCCNF significantly. Remarkably, with relatively small training samples (with threshold $40\%$), our model again superior to all generators, demonstrating that our model achieves consistently good and stable generation results even with limited data.

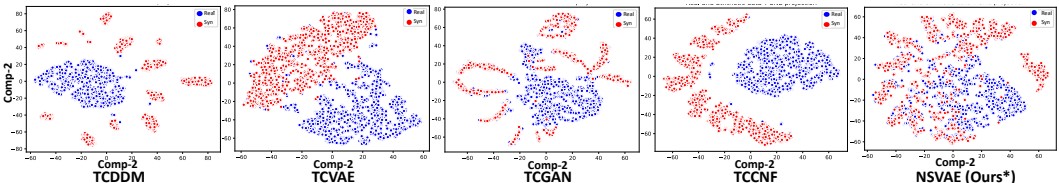

Figure 2: t-SNE plots for our proposed model on MIMIC-IV dataset using $100\%$ training data. Blue is for original data, and Red for synthetic data.

**Generative Performance Evaluated by t-SNE Charts**  In Fig.2, the t-SNE plots depict data generated by our model for the MIMIC-IV dataset using a training threshold of $100\%$. Our model's generated data exhibits significant overlap with the original data, contrasting sharply with the noisy outputs from other generators, thus further highlighting the superior generative performance of our model.

### 5.4 NEURO-SYMBOLIC FORWARD REASONING

In Fig.3, we visualize the process of neuro-symbolic forward reasoning for the experiment on MIMIC-IV dataset. We start from an initial guess of latent variable $z$, which indicate each high-level concept's satisfaction. From the heatmap, one can see that the satisfaction of Concept-1: Abnormal blood pressure and blood oxygen saturation progressively increase, indicating that in the forward reasoning process, the significance of this concept gradually amplifies, with our neuro-symbolic layer inferring its pivotal role in the original data distribution. Similar patterns can be further affirmed from the rules mined from the neuro-symbolic layer as shown in in Appendix.E. This concept appears in 4 out of all 5 mined rules. The satisfaction of Concept-4: Electrolyte Imbalance and Concept-9: Abnormal Urine Output also have been enhanced. These results demonstrate the stable reasoning capacity of our proposed model.

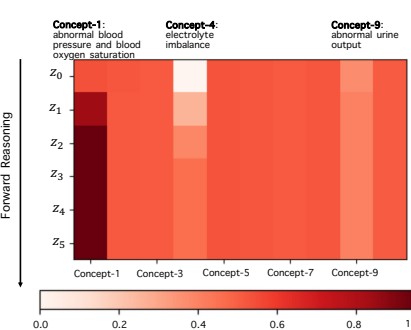

Figure 3: Change of inferred $z$ during the process of neuro-symbolic forward chaining on MIMIC-IV dataset

## 6 CONCLUSION

We propose a novel VAE framework that integrates a neural-symbolic reasoning layer into the latent space, enabling the incorporation of interpretable, logic-based constraints and logical reasoning on learned representations. Our model addresses the complex task of temporal point process generation, crucial for generating de-identified data and managing missing data. Proficient models can utilize domain expertise to produce semi-synthetic datasets from real data, facilitating transfer learning and ensuring secure data generation for privacy protection.

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

## Appendix Overview

In the following, we will provide supplementary materials to better illustrate our methods and experiments.

- Section.A provides detailed datasets introduction and preprocessing methods.
- Section.B provides the feature definition and corresponding high-level concepts for all the real-world datasets.
- Section.C comprehensively introduces the baseline methods we considered in our paper.
- Section.D reports the details of experiments for prediction tasks.
- Section.E reports the learned rules via neuro-symbolic forward reasoning on MIMIC-IV dataset
- Section.F record the time efficiency of our proposed method.
- Section.G provides the information of computing infrastructure for all experiments.

## A  Datasets Details

We extracted four interesting real-world datasets. Followings are brief introduction to these real-world datasets: *i) MIMIC-IV*: an electronic health record dataset of patients admitted to the intensive care unit (ICU). (Johnson et al., 2023). We considered patients diagnosed with sepsis (Saria, 2018), one of the major causes of mortality in ICU due to septic shock. We extract 2000 samples of multiple features with average sequence length of 22.93 events, encompassing lab measurements, drug intake, intravenous fluids, and urine output. *ii) Covid-19 UK*: COVID-19 is an unprecedented pandemic and various control measures have been introduced to curb the spread of the virus. The Oxford Covid-19 Government Response Tracker (OxCGRT) gathers data on governments' implementation of specific measures and their timing. (Hale et al., 2021; 2020). We collected 27 samples, each with sequence length of 59.22 events, for the United Kingdom during 2021, focusing on the effect of government's epidemic prevention policies related to containment/closure, the healthcare system, vaccination efforts, and economic impacts on daily cumulative number of confirmed cases reduction. To reduce daily fluctuations, we recorded the cumulative number of confirmed cases over 7-day intervals to illustrate the epidemic spread trend. We identified the time points when case numbers began to decrease. *iii) Car-Follow*: a dataset processed from Lyft level-5 open dataset (Li et al., 2023; Houston et al., 2021), which includes 1000+ hours of perception and motion data collected over a 4-month period from urban and suburban environments along a fixed route in Palo Alto, California. We extract 5000 samples with an average sequence length of 4.6 events, which recordings vehicle driving modes. *iv) Epic-Kitchen*: This dataset originates from a large-scale, first-person (egocentric) vision dataset, featuring multi-faceted, audio-visual, non-scripted recordings in natural settings, specifically the wearers' homes. It captures daily kitchen activities over multiple days. We have utilized the annotated action sequences. focusing only text, and extracted them to create a temporal event history of cooking verbs. This was achieved by omitting the entities that the human subjects interacted with. We specifically focus on ten verbs such as manipulate, move, clean, etc. We concentrated on a subset of 400 samples, each with an average sequence length of 36.76 events.

## B  Features and High-Level Concepts for Datasets

## C  Baselines

In this paper, we primarily focus on baselines from three different fields: neural Temporal Point Process model, Logic-Based model, and generative model. Below, we will provide a detailed introduction to these baselines.

- **Neural Temporal Point Process Model**
  - RMTPP (Du et al., 2016): The approach considers the intensity function of a temporal point process as a nonlinear function that depends on the history. It utilizes a recurrent

| Concept Number | Concept Content | Predicates |
|---|---|---|
| Concept-1 | Abnormal blood pressure and blood oxygen saturation | Abnormal-SpO2SaO2 |
| Concept-2 | Abnormal blood volume | Abnormal-CVP |
| Concept-3 | Abnormal vascular resistance | Abnormal-SVR |
| Concept-4 | Electrolyte imbalance | Abnormal-Potassium
Abnormal-Sodium
Abnormal-Chloride |
| Concept-5 | Abnormal kidney function markers | Abnormal-BUN
Abnormal-Creatinine |
| Concept-6 | Abnormal inflammatory markers | Abnormal-CRP |
| Concept-7 | Abnormal blood cell counts | Abnormal-RBCcount
Abnormal-WBCcount |
| Concept-8 | Abnormal blood gas analysis | Abnormal-ArterialpH
Abnormal-ArterialBE
Abnormal-Lactete
Abnormal-HCO3
Abnormal-SvO2ScvO2 |
| Concept-9 | Abnormal urine output | Low-Urine |
| Concept-10 | Use drug | Colloid
Crystalloid
Water
Norepinephrine
Epinephrine
Dobutamine
Dopamine
Phenylephrine |

Table 3: Defined predicates and corresponding high-level concepts for MIMIC-IV dataset.

| Concept Number | Concept Content | Predicates |
|---|---|---|
| Concept-1 | Containment and closure policies | School Closing
Workplace Closing
Cancel Public Events
Restrictions on Gathering Size
Close Public Transport
Stay at Home Requirements
Restrictions on Internal Movement
Restrictions on International Travel |
| Concept-2 | Vaccination policies | Vaccine Prioritisation
Vaccine Eligibility/Availability
Vaccine Financial Support
Mandatory Vaccination |
| Concept-3 | Health system policies | Public Information Campaign
Testing Policy
Contact Tracing |
| Concept-4 | Economic policies | Income Support
Debt/Contract Relief for Households |
| Concept-5 | Effective policy | Cumulative Confirmed Cases Decrease |

Table 4: Defined predicates and corresponding high-level concepts for Covid-19 UK dataset.

neural network to automatically learn a representation of the influences from the event history, which includes past events and time intervals, thereby fitting the intensity function of the temporal point process.

- THP (Zuo et al., 2020): The model employs a concurrent self-attention module to embed historical events and generate hidden representations for discrete time stamps. These hidden representations are then used to model the interpolated continuous time

| Concept Number | Concept Content | Predicates |
|---|---|---|
| Concept-1 | Aggressive action | Acceleration Following a Leading Vehicle
Free Acceleration |
| Concept-2 | Conservative action | Deceleration Following a Leading Vehicle |
| Concept-3 | Nnormal action | Cruising at a Desired Speed
Constant Speed Following |

Table 5: Defined predicates and corresponding high-level concepts for Car-Following dataset.

| Concept Number | Concept Content | Predicates |
|---|---|---|
| Concept-1 | Manipulation | Manipulate
Control |
| Concept-2 | Food-Handling | Mix and Stir
Clean
Food Handling |
| Concept-3 | Movement | Move |
| Concept-4 | Organization | Organize
Retrieve |
| Concept-5 | Inspection | Inspect |
| Concept-6 | Miscellaneous | Miscellaneous |

Table 6: Defined predicates and corresponding high-level concepts for Epic-Kitchen-100 dataset.

intensity function. THP can also incorporate additional structural knowledge. Importantly, THP surpasses RNN-based approaches in terms of computational efficiency and the ability to capture long-term dependencies.

- PromptTPP (Xue et al., 2023): The model incorporates a continuous-time retrieval prompt pool into the base TPP, enabling sequential learning of event streams without the need for buffering past examples or task-specific attributes. Specifically, this approach consists of a base TPP model, a pool of continuous-time retrieval prompts, and a prompt-event interaction layer. By addressing the challenges associated with modeling streaming event sequences, this mode enhances the model's performance.

- HYPRO (Xue et al., 2022): The hybridly normalized probabilistic (HYPRO) model is capable of making long-horizon predictions for event sequences. This model consists of two modules: the first module is an auto-regressive base TPP model that generates prediction proposals, while the second module is an energy function that assigns weights to the proposals, prioritizing more realistic predictions with higher probabilities. This design effectively mitigates the cascading errors commonly experienced by auto-regressive TPP models in prediction tasks, thereby improving the model's accuracy in long-term forecasting.

- **Logic-Based Model**

  - TELLER (Li et al., 2021): It is a non-differentiable algorithm that can be described as a temporal logic rule learning algorithm based on column generation principles. This method formulates the process of discovering rules from noisy event data as a maximum likelihood problem. It also designs a tractable branch-and-price algorithm to systematically search for new rules and expand existing ones. The algorithm alternates between a rule generation stage and a rule evaluation stage, gradually uncovering the most significant set of logic rules within a predefined time limit.

  - CLNN (Yan et al., 2023): The model learns weighted clock logic (wCL) formulas, which serve as interpretable temporal logic rules indicating how certain events can promote or inhibit others. Specifically, the CLNN model captures temporal relations between events through conditional intensity rates guided by a set of wCL formulas that offer greater expressiveness. In contrast to conventional approaches that rely on computationally expensive combinatorial optimization to search for generative rules, CLNN employs smooth activation functions for the components of wCL formulas. This enables a continuous relaxation of the discrete search space and facilitates efficient learning of wCL formulas using gradient-based methods.

- **Generative Model**: All the temporal point process generative models we consider in our paper are summarized in the work of (Lin et al., 2022). It simplifies the generative model of temporal point processes into an history-encoder-probabilistic-decoder architecture. For the history encoder, we use attention mechanism Vaswani (2017); Zuo et al. (2020). For the generative probabilistic decoder, we consider

  - TCDDM (Sohl-Dickstein et al., 2015): Temporal conditional diffusion denoising model (TCDDM) is based on diffusion model. In sampling, given the historical encoding, we first sample from the standard normal distribution, then take it and historical encoding as the input to get the approximated noise, and generally remove the noise with different scales to recover the samples. For inference, the prediction is based on Monte Carlo estimation.
  - TCVAE (Kingma, 2013; Pan et al., 2020): Temporal conditional variational autoencoder (TCVAE) consists of a variational encoder as a conditional Gaussian distribution for approximating the prior standard Gaussian and a variational decoder to generate arrival time samples.
  - TCGAN (Xiao et al., 2017a): Temporal conditional generative adversarial network (TCGAN) decoder is mostly based on Wasserstein GAN in TPPs (Arjovsky et al., 2017; Xiao et al., 2017a). The probabilistic generator is trained via adversarial process, in which the other network called discriminator is trained to map the samples to a scalar, for maximizing the Wasserstein distance between the distribution of generated samples and the distribution of observed samples.
  - TCCNF (Mehrasa et al., 2019): Temporal conditional continuous normalizing flows (TCCNF) is based on Neural ODE (Chen et al., 2018; 2020).

## D    DETAILS OF EXPERIMENTS FOR PREDICTION TASKS

In Tab. 7, Tab.8, Tab.9, and Tab.10 we present the mean $ER\%$ and $MAE$ across four datasets for various baselines, averaged over three separate seed experiments, along with their respective standard deviations. Our method consistently outperforms all baseline models across these datasets.

| Category | Model | MIMIC-IV | |
|---|---|---|---|
| | | ER% $\downarrow$ | MAE $\downarrow$ |
| Neural TPP | RMTPP | 92.12% +/- 1.25% | 3.75 +/- 0.25 |
| | THP | 90.38% +/- 1.25% | 3.52 +/- 0.33 |
| | PromptTPP | 86.23% +/- 1.50% | 3.27 +/- 0.23 |
| | HYPRO | 86.87% +/- 2.46% | 3.20 +/- 0.15 |
| Logic Model | TELLER | 88.85% +/- 1.86% | 3.54 +/- 0.59 |
| | CLNN | 87.43% +/- 1.43% | 3.48 +/- 0.42 |
| Gen. Model | TCDDM | 87.58% +/- 8.66% | 3.36 +/- 0.35 |
| | TCVAE | 86.67% +/- 7.01% | 3.40 +/- 0.29 |
| | TCGAN | 85.97% +/- 5.30% | 3.29 +/- 0.46 |
| | TCCNF | 91.20% +/- 3.63% | 3.76 +/- 0.70 |
| | **Ours\*** | **85.59% +/- 2.75%** | **3.13 +/- 0.20** |

Table 7: Comparison between our model and baselines for prediction tasks on MIMIC-IV dataset. Bold text represents the best result. The performance is averaged over three different seeds and the standard deviation is stored after "+/-".

## E    LEARNED RULES

The learned rule can be found in Tab.11.

## F    TIME EFFICIENCY

We record the training time for all the generative model using 100% training data. Results shown in Tab.12 indicate that our proposed model requires significantly less computing time.

| Category | Model | Covid-19 UK | |
| --- | --- | --- | --- |
| | | ER% ↓ | MAE ↓ |
| Neural TPP | RMTPP | 62.57% +/- 1.45% | 3.52 +/- 0.43 |
| | THP | 60.74% +/- 1.50% | 3.20 +/- 0.33 |
| | PromptTPP | 54.80% +/- 2.68% | 2.95 +/- 0.10 |
| | HYPRO | **49.10% +/- 1.75%** | **2.58 +/- 0.21** |
| Logic Model | TELLER | 58.90% +/- 7.28% | 3.02 +/- 0.23 |
| | CLNN | 57.86% +/- 6.26% | 2.87 +/- 0.02 |
| Gen. Model | TCDDM | 58.23% +/- 5.28% | 3.31 +/- 1.23 |
| | TCVAE | 59.34% +/- 6.23% | 3.02 +/- 0.63 |
| | TCGAN | 58.02% +/- 4.23% | 3.12 +/- 0.35 |
| | TCCNF | 60.10% +/- 9.48% | 3.25 +/- 0.87 |
| | **Ours*** | 53.68% +/- 0.83% | 2.74 +/- 0.08 |

Table 8: Comparison between our model and baselines for prediction tasks on Covid-19 UK dataset. Bold text represents the best result. The performance is averaged over three different seeds and the standard deviation is stored after "+/-".

| Category | Model | Car Following | |
| --- | --- | --- | --- |
| | | ER% ↓ | MAE ↓ |
| Neural TPP | RMTPP | 36.27% +/- 2.57% | 2.64 +/- 0.23 |
| | THP | 34.70% +/- 4.39% | 2.30 +/- 0.20 |
| | PromptTPP | 34.56% +/- 1.23% | 2.10 +/- 0.10 |
| | HYPRO | 34.35% +/- 1.03% | 2.23 +/- 0.32 |
| Logic Model | TELLER | 40.25% +/- 5.23% | 3.41 +/- 0.50 |
| | CLNN | 39.75% +/- 4.25% | 3.35 +/- 0.33 |
| Gen. Model | TCDDM | 35.38% +/- 6.23% | 2.32 +/- 0.32 |
| | TCVAE | 37.76% +/- 3.00% | 2.48 +/- 0.82 |
| | TCGAN | 34.20% +/- 2.54% | 2.58 +/- 0.66 |
| | TCCNF | 40.29% +/- 7.66% | 2.80 +/- 1.02 |
| | **Ours*** | **33.26% +/- 2.00%** | **1.92 +/- 0.15** |

Table 9: Comparison between our model and baselines for prediction tasks on Car Following dataset. Bold text represents the best result. The performance is averaged over three different seeds and the standard deviation is stored after "+/-".

## G  COMPUTING INFRASTRUCTURE

All synthetic data experiments, as well as the real-world data experiments, including the comparison experiments with baselines, are performed on Ubuntu 20.04.3 LTS system with Intel(R) Xeon(R) Gold 6248R CPU @ 3.00GHz, 227 Gigabyte memory.

| Category | Model | Epic-Kitchen | |
| --- | --- | --- | --- |
| | | ER% ↓ | MAE ↓ |
| Neural TPP | RMTPP | 42.84% +/- 4.28% | 9.21 +/- 2.37 |
| | THP | 40.25% +/- 4.62% | 9.05 +/- 2.56 |
| | PromptTPP | 37.50% +/- 3.62% | 7.80 +/- 1.80 |
| | HYPRO | 38.25% +/- 3.64% | 8.12 +/- 2.00 |
| Logic Model | TELLER | 41.23% +/- 4.21% | 8.83 +/- 3.25 |
| | CLNN | 40.85% +/- 3.64% | 8.30 +/- 3.19 |
| Gen. Model | TCDDM | 45.34% +/- 6.32% | 8.34 +/- 3.54 |
| | TCVAE | 37.10% +/- 5.27% | 7.87 +/- 4.25 |
| | TCGAN | 39.83% +/- 3.66% | 8.20 +/- 2.25 |
| | TCCNF | 46.83% +/- 7.94% | 9.28 +/- 4.63 |
| | **Ours*** | **36.16**% **+/- 2.25**% | **7.20 +/- 0.85** |

Table 10: Comparison between our model and baselines for prediction tasks on Epic-Kitchen 100 dataset. Bold text represents the best result. The performance is averaged over three different seeds and the standard deviation is stored after "+/-".

| Learned Rules |
| --- |
| **Rule-1**: Abnormal urine output ← Abnormal blood pressure and blood oxygen saturation ∧ Abnormal inflammation markers |
| **Rule-2**: Use Drug ← Abnormal blood pressure and blood oxygen saturation ∧ Abnormal urine urine output |
| **Rule-3**: Abnormal blood volume ← Abnormal blood pressure and blood oxygen saturation ∧ Abnormal blood cell counts ∧ Abnormal urine output |
| **Rule-4**: Electrolyte imbalance ← Abnormal real-time urine output ∧ Use drug |
| **Rule-5**: Abnormal kidney function markers ← Abnormal blood pressure and blood oxygen saturation ∧ Abnormal blood volume ∧ Abnormal vascular resistance |

Table 11: Learned rules via neuro-symbolic forward reasoning on MIMIC-IV dataset

| Model | MIMIC-IC | Covid-19 | Car-Following | EPIC-Kitchen |
| --- | --- | --- | --- | --- |
| TCDDM | 16448.49 | 12540.34 | 9547.85 | 10032.49 |
| TCVAE | 2471.40 | 2018.78 | 1479.45 | 1885.20 |
| TCGAN | 29234.03 | 19656.21 | 23004.83 | 25690.34 |
| TCCNF | 26604.10 | 11448.90 | 18432.47 | 23580.23 |
| **Ours*** | 1624.65 | 1075.22 | 921.60 | 1367.16 |

Table 12: Training times (in seconds) for all generative models using 100% of the training data. Times were all obtained using the same computation infrastructure, which can be found in Appendix.G.

