# OpenReview forum: "Variational Neuro-Symbolic Generative Temporal Point Process"
_ICLR.cc/2025/Conference — ICLR 2025 Conference Withdrawn Submission_

### Official Review · Reviewer_Hao8 · 2024-10-29

**Soundness:** 1
**Presentation:** 3
**Contribution:** 2
**Rating:** 3
**Confidence:** 4

**Summary:**

This work aims at integrating neural-symbolic reasoning and representation with variational auto encoders for the purposes of automatically learning discrete, generative rules for (marked) temporal point processes. This is done by first defining pre-specified concepts / predicates satisfied by events in a sequence, that are then embedded along with the sequence itself, and then used to decode a sequence-specific rule or clause $z$. Sequence events are then autoregressively modeled while conditioning on the decoded rule $z$. The model is trained to maximize the ELBO, and was examined empirically on its predictive and generative performance, along with some qualitative examinations of the identified rules.

**Strengths:**

The general idea of learning interpretable rules for event sequence generation is very interesting and a promising future for the TPP space in general as modern-day models become more and more black-box. Additionally, being able to incorporate expert-specified rules and concepts is very useful and allows for various use-cases that would not be available otherwise. As for the methodology used, the architecture and approach used does appear to be conducive towards amortizing rule estimation on a per-sequence basis. This is important as many datasets can consist of multiple different types of sequences, generated by potentially different sources (e.g., users) that do not utilize the exact same marginal rule set.

**Weaknesses:**

The is one critical weakness in this paper that ultimately led to my “poor” soundness score and overall rating. The proposed method is repeatedly claim superior predictive and generative performance with respect to modeling event sequences when compared to, among others, more traditional neural TPP models; however, as far as I can tell all of these results are invalid comparisons. This is because of how predictions and generations are being made with the proposed approach. This process is only mentioned in passing once, without any alternative mentioned, on line 358: “In the generation phase, after the model being well-trained, we start from an initial state and sample $z$ from $\hat{z}^{(H)}$…” For both prediction and generation tasks this implies that there is a data leakage as $\hat{z}^{(H)}$ is the result of encoding the very sequence being compared against. When performing a sequential VAE, there are generally two high-level approaches to take:

1. Encode the entire sequence, $z = e(x_1, \dots, x_n)$, and then decode the entire sequence $\hat{x}_1, \dots, \hat{x}_n = d(z)$ (could be done auto regressively, or
2. Encode partial histories of the sequence, $z_i = e(x_1, \dots, x_i)$ for $I=1,…,n$, and then partially decode continuations of the sequence, $\hat{x}_i = d(z_i, x_1, \dots, x_{i-1})$.

This work clearly falls under the first scenario, and because of this during prediction to not ‘cheat’ one must sample $z$ from a prior rather than the approximate posterior $q(z | x)$. The take-away from this is to either convert the setup into something more resembling (2), which can iteratively adapt the hidden state as new information comes in, or remain as (1) but put much more focus on the interpretation of the posterior $q(z | x)$. From what I can tell, this is the one valid thing the paper did do with regards to empirical results as there is no data leakage possible here. That being said, there is not enough investigation into this part of the model (as far as I can tell, a small paragraph in Sec 5.4 and a single Table 11) to justify overlooking the prediction and generation issues.

**Questions:**

Please address the main weakness outlined above. Additionally, I do have a few questions that seek clarification regarding some details.

It is my understanding that the $d$ concepts are specified by experts ahead of time and that their embeddings $\theta_i$ for $I=1,\dots,d$ can be learned via pre-training ahead of time. My first question is, for the results shown in the paper is this pre-training done? If so, what are the details concerning this? I understand the paper goes on to later say that “we directly learn the encoder to align with latent predicates…” and that “This design removes the reliance on pre-training and allows the model to learn the latent concepts (predicates) dynamically based on data alignment during training.” This leads me to believe that no pertaining is done; however, should this be the case then that leads me to my next question which is how the semantic meaning of these expert-specified concepts is enforced? If there is no supervised model using explicit labels, then what is to say that for $\theta_2$ really represents “Abnormal blood volume” in MIMIC-IV and not some other concept, either specified or just abstractly learned? Please let me know if I missed some key details regarding this.

Lastly, for the prior $p(z)$ what are the prior probabilities $p_i$ chosen to be? Do they depend on the dataset / concepts specified? How do the results change as this hyper parameter change?

---

### Official Review · Reviewer_RS22 · 2024-11-03

**Soundness:** 2
**Presentation:** 3
**Contribution:** 3
**Rating:** 5
**Confidence:** 3

**Summary:**

This research paper presents a new approach to generating synthetic event sequences based on Temporal Point Processes (TPPs) especially for application in health such as electronic health records (EHRs). The paper proposes a Variational Autoencoder (VAE) model with a neuro-symbolic reasoning layer (by forward chaining) that allows the integration of interpretable logic-based constraints, thereby improving the accuracy, interpretability, and trustworthiness of the generated sequences. The authors test their model on various real-world datasets demonstrating its effectiveness in capturing complex event dynamics while adhering to clinical logic and generating clinically relevant synthetic data.

**Strengths:**

Originality: this is a novel generative model for temporal point processes although it is combination of older ideas VAE + logic TPP (by Shuang Li et al, icml 2020).

Quality: the overall quality is good; it introduces relative background, and then close some gap between current state of the art models to application specific domain by proposing the neuro symbolic VAE approach, and it demonstrate the effective by prediction + generation, and with visualization and real examples.

Clarity: it is well presented for the most part.

Significance: I think this is an interest line of research and meaningful within clinical TPP modeling, to some extent, however there are a few other closely related work ( such as logic TPPs by Li et al, rule induction by Yang et al. )

**Weaknesses:**

My main concern is the updating of latent variable z for predicates or concepts. The authors just mechanistically describe how they update and leverage forward chain, however I am not sure how the updates are grounded. After applying to H iteration, does z reach some equilibrium?  It would be helpful for the authors to provide more details on the theoretical or empirical justification for their update process.

A minor is the authors should briefly discuss how this work differs from previous yang et al, from line 113-114.

**Questions:**

1 Can the authors clarify the difference between V and theta? Did authors need to pretrain to learn theta’s? The authors also assume V_f has a fix dimension, and L being the length of predicates including body and head. However this fixed length has limitation in practice since not all rules have the same fixed length. How can this handled?

2. Are eqn 2 and 6 are meant to be identical? if not, please clarify the differences between them and explain the significance of those differences.

3. The authors propose to evaluate the  generation task discriminatively and the performance of the model is determined by the classifier. What classifier is used? Also can the author think of another metric that measure sequence distances such as maximum mean discrepancy MMD?

---

### Official Review · Reviewer_zg9C · 2024-11-04

**Soundness:** 3
**Presentation:** 3
**Contribution:** 3
**Rating:** 6
**Confidence:** 3

**Summary:**

This paper presents a neuro-symbolic generative model for temporal point processes (TPPs) based on VAE framework. The authors incorporate a neural-symbolic reasoning layer into the latent space to integrate interpretable logic-based constraints. The main motivation is to improve the interpretability of the latent representation.

**Strengths:**

* The authors do a lot of engineering to justify their architectural choice, especially for their generative decoder.
* The comparison with existing models are pretty comprehensive, covering both Neural TPP and Logic models.
* The interpretability of binary hidden representation and forward reasoning is well protrayed.

**Weaknesses:**

* The method bears much similarity to Neuro-Symbolic Temporal Point Processes (Yang 2024), especially consider both use predicate embeddings, represent rules as vector embeddings, and use similarity measures for rule matching. The main difference is the additional binary representation by VAE and Yang et al. uses greedy optimization (per rule) instead of joint optimization of all rules. However, the comparison with Yang et al. is negligible.
* Using VAE for TPP is not new (we have too many VAE-based TPPs); the motivation for VAE here is unclear. It seems like your reasoning process only needs a latent variable (z in your case), so why do we need an additional sampling step? We should be able to directly send encoded x to the reasoning process. The previous work does not need a VAE. The TPP is already stochastic, so you are essentially introducing extra stochasticity without obvious reasons.
* The interpetability here is more or less a "sanity check". Seeing concept satisfaction increases over time is good but cannot really help user make any judgments — we only know that the model is making good use of the concept. When using neural sympolic model, common motivations are checking more straightforward interactions between rules and data, but in your paper it seems such interaction is more much complicated than that of Yang et. al.

**Questions:**

* I understand Yang 2024 is a very recent paper, but do you have a valid reason for not comparing with it? e.g. lack of public codebase?
* It seems that the related works are not very polished.

> We aim to utilize TPP to model the real-world event sequences.

I think pervious works also apply to real-world event sequences.

> However, existing VAE frameworks often lack interpretability in the latent representation, overlooking fine-grained guiding logic rules.

Isn't that normal? People use VAE trade-off interpretability for flexibility.

* What is your justification for adding VAE (making the process doubly stochastic)? Your experiments seem not covering the no-VAE case. Again, without VAE, TPP is still a gen model (like LLM does not need a sampling step).
* What are the supposed use case of your model's interpretability?
* What is the computational complexity of the forward reasoning process?

---

### Official Review · Reviewer_aeYj · 2024-11-09

**Soundness:** 2
**Presentation:** 3
**Contribution:** 2
**Rating:** 3
**Confidence:** 4

**Summary:**

The authors propose a VAE for temporal event sequences in which the latent representation is comprised of binary variables indicating whether corresponding logical predicates are satisfied. Some predicates are primitives that are pre-defined, while others are derived according to learned rules. The authors' goal is to make the representations more interpretable while also improving prediction and data generation quality. The authors compare prediction and generation performance with baseline methods (TPPs, logical models, generative models) across four datasets and provide examples of learned rules.

**Strengths:**

- Prediction and generation from interpretable predicates would be beneficial in medical and other settings
- The background is comprehensive and clearly presented
- The mathematical presentation and methodological details are easy to follow
- Experimental settings are comprehensive, and the evaluation is multi-faceted
- Results show benefit in most of the experimental settings / configurations

**Weaknesses:**

Descriptions of the concepts and how they are extracted are underspecified. The authors say that some are pre-specified (line 236) and others can be learned, and they say that they "construct ground truth governing logic rules either by experts or LLMs, with details illustrated in Appendix B". But Appendix B contains only a limited list of concepts, and no details on how pre-specified concepts are chosen or extracted is provided.

Other methodological details are similarly unclear. For example, I could not find details about the size of the latent representation space used in the experiments. It is hard for me to believe that the very limited number of concepts and rules described in the Appendix would be sufficient to achieve good reconstruction performance, so I suspect that the latent representations are high-dimensional. However, if this is the case, it undercuts the interpretability of the method, which is a key advantage noted by the authors.

The concepts and rules presented in the Appendix don't make much sense, and it is not clear to me how they were defined or whether the lists presented are comprehensive. Is the Appendix presenting only a subset of concepts that were cohesive and easy to interpret? And if so, is the method truly interpretable?

The results suggest that the proposed method consistently yields better predictions and more realistic data while still being more interpretable compared to a wide range of established methods. This seems too good to be true, and I don't understand why it would be the case. I understand and agree with the benefits to interpretability, but the authors provide no explanation as to why the approach ought to improve performance compared to alternatives, which makes me suspect that results may have been cherry-picked or other methods may not have been tuned adequately.

**Questions:**

- What is the dimensionality of the representation space in the experiments?
- What is the length L (max number of predicates per rule) in the experiments?
- What are the implications of the multiple forward passes for model learning?
- Why would prediction and generation performance be better with this approach?
- How are the pre-specified concepts initially chosen and extracted?

---

### Note · Authors · 2024-11-25

**Comment:**

We sincerely appreciate the time and effort the reviewers, PCs, SACs, and ACs  have taken to provide such thoughtful and detailed feedback on our work. We will further improve our paper.

**Withdrawal Confirmation:**

I have read and agree with the venue's withdrawal policy on behalf of myself and my co-authors.